# Overexpression of 9-*cis*-Epoxycarotenoid Dioxygenase Gene, *IbNCED1*, Negatively Regulates Plant Height in Transgenic Sweet Potato

**DOI:** 10.3390/ijms241310421

**Published:** 2023-06-21

**Authors:** Yuanyuan Zhou, Chunling Zhao, Taifeng Du, Aixian Li, Zhen Qin, Liming Zhang, Shunxu Dong, Qingmei Wang, Fuyun Hou

**Affiliations:** Crop Research Institute, Shandong Academy of Agricultural Sciences, Jinan 250100, China; zhou_yy_2020@163.com (Y.Z.);

**Keywords:** sweet potato, *IbNCED1*, plant height, ABA, GA

## Abstract

Plant height is one of the key agronomic traits for improving the yield of sweet potato. Phytohormones, especially gibberellins (GAs), are crucial to regulate plant height. The enzyme 9-*cis*-epoxycarotenoid dioxygenase (NCED) is the key enzyme for abscisic acid (ABA) biosynthesis signalling in higher plants. However, its role in regulating plant height has not been reported to date. Here, we cloned a new NCED gene, *IbNCED1*, from the sweet potato cultivar Jishu26. This gene encoded the 587-amino acid polypeptide containing an NCED superfamily domain. The expression level of *IbNCED1* was highest in the stem and the old tissues in the in vitro-grown and field-grown Jishu26, respectively. The expression of *IbNCED1* was induced by ABA and GA3. Overexpression of *IbNCED1* promoted the accumulation of ABA and inhibited the content of active GA3 and plant height and affected the expression levels of genes involved in the GA metabolic pathway. Exogenous application of GA3 could rescue the dwarf phenotype. In conclusion, we suggest that *IbNCED1* regulates plant height and development by controlling the ABA and GA signalling pathways in transgenic sweet potato.

## 1. Introduction

Sweet potato, *Ipomoea batatas* (L.) Lam., is an important root crop worldwide [1,2]. In actual production, due to the genotype, excessive nitrogen application, uneven rainfall distribution, and improper irrigation, sweet potato is easily overgrown; this seriously impacts the yield, mechanization degree and the sustainable development of the sweet potato industry [3]. The ideal plant height of sweet potato helps to break through the bottleneck. However, research to date has yet to establish a clear genetic basis and the constituent elements in sweet potato.

Plant height is an agronomic trait with a complex genetic basis [4,5]. It is confined by stem elongation and plays important role in crop yield and quality [6]. With the rise of the green revolution, a large number of dwarf mutants, quantitative trait loci (QTLs), and genes have been identified to control plant height [7,8,9,10]. In wheat, the Reduced height (Rht) alleles, such as *Rht-1*, *Rht-B1b*, and *Rht-D1b* were introduced to reduce plant height, providing improved lodging resistance through interfering with the action or production of the gibberellins (GAs), plant hormones [11,12,13].

GAs are a class of tetracyclic diterpenoid phytohormones that mediate different processes of plant development including stem elongation, seed germination, trichome development, leaf expansion, induction of flowering, and pollen maturation [14,15]. More than 130 GAs have been identified, and GA1, GA3, GA4, and GA7 show capital biological activity that controls plant development [16,17,18]. Higher GA levels and more active GA biosynthesis were found to be correlated with plant height [19,20,21]. GA metabolism or signalling conferred grain productivity during the Green Revolution by reshaping plant stature [22,23]. Many genes have been identified relating to plant height through the GA signalling pathway. In rice, *OsDREB2B*, *OsAP2-39*, and *OsWRKY21* reduce plant height development through the GA biosynthesis pathway [24]. TaLecRK-IV.1 and TaRht24 are regulators of plant height through the gibberellic acid and auxin-signalling pathways in wheat [25,26]. Overexpression of *CmDRP* resulted in a semi-dwarf phenotype with a significantly decreased active GA3 content, while reduced expression generated the opposite phenotype inchrysanthemum [27].

It has always been clear that GAs interact with other plant hormones [28]. GA and ABA usually play antagonistic roles in the regulation of germination, growth, and flowering in plants [29,30]. ABA affect the GA pathway through a different mechanism, such as an ABA-induced Ser/Thr protein kinase (PKABA1) and transcriptional regulators of ABA-induced WRKY, DELLA, and MYB [31,32,33,34,35]. *Arabidopsis* ABF2 and ABF4 transcription factors positively regulate potato tuber induction by regulating the expression of ABA- and GA-metabolism genes [36].

The enzyme 9-*cis*-epoxycarotenoid dioxygenase (NCED) is the key enzyme for ABA biosynthesis signalling [37,38]. NCED genes are associated with development and tolerance by the ABA signalling pathway in plants. Overexpression of *VaNCED1* delayed the development of transgenic *Vitis vinifera* [39]. *OsNCED3* and *OsNCED5* mediated seed dormancy, plant growth, abiotic stress tolerance, and leaf senescence by regulating ABA biosynthesis in rice [40,41]. *LeNCED1* overexpression in tomato increased ABA concentration and prevented the induction of genes involved in ABA metabolism and the deactivations of GA and auxin that occurred in WT [42]. The expression of NCED genes in dwarf cotton accession was higher than that in taller ones, and *GhNCED1*-silenced cotton plants increased in height [43]. Up to now, NCED genes have not been identified in sweet potato. In this study, we cloned a new *IbNCED1* gene for the 587aa from sweet potato. Functional analysis showed that *IbNCED1* enhanced the accumulation of ABA and inhibited plant height, affected the expression levels of genes involved in the GA metabolic pathway, and affected the content of active GA.

## 2. Results

### 2.1. Cloning and Sequence Analysis of IbNCED1

The novel *IbNCED1* gene was isolated from the sweet potato cultivar Jishu26. The 1764-bp ORF sequence of *IbNCED1* encoded a protein of 587 aa with a molecular weight of 65.33 kDa and a predicted *p*I of 6.12, which belongs to the RPE65 superfamily (Figure 1A). Phylogenetic analysis of NCED proteins with a neighbor-joining method revealed that *IbNCED1* has high homology with NCED proteins from *Ipomoea triloba* (*ItNCED1*, XP_031110150.1), *Ipomoea nil* (*InNCED1*, XP_019153780.1), and *Solanum lycopersicum* (*SlNCED1*, NP_001234455.1) (Figure 1B).

### 2.2. Expression Analysis of IbNCED1

To study the potential function of *IbNCED1* in sweet potato, its expression in different tissues and treatments of Jishu26 was analyzed with qRT-PCR. The expression level of *IbNCED1* was the highest in the stem of the in vitro-grown Jishu26 plants (Figure 2A). For the field-grown Jishu26 plants, the expression level of *IbNCED1* was higher in the old stem, pencil root, and storage root tissues than in other young tissues (Figure 2B).

The expression of *IbNCED1* was downregulated in the leaf and upregulated in the stem and root after ABA and GA treatments. The expression level peaked at 3 h (4.520-fold in the stem and 4.56-fold in the root) after ABA treatment (Figure 2C), while it peaked at 12 h in the stem and at 6 h in the root (4.11- and 3.08-fold, respectively) after GA treatment (Figure 2D). These results suggest that *IbNCED1* might be involved in ABA and GA response pathways.

### 2.3. Regeneration of the Transgenic Sweet Potato Plants

The overexpression vector pCAMBIA1301s-*IbNCED1* was introduced into the *Agrobacterium tumefaciens* strain EHA105 (Figure 3A). Cell aggregates of Xushu22 (Figure 3B) co-cultivated with EHA105 carrying pCAMBIA1301-*IbNCED1* were cultured on the selective MS medium with 2.0 mg L^−1^ 2,4-dichlorophenoxyacetic acid (2,4-D), 100 mg L^−1^ carbenicillin (Carb), and 10 mg L^−1^ hygromycin (Hyg) (Figure 3C). Seventeen Hyg-resistant embryogenic calluses of 132 cell aggregates were obtained after 6 weeks. These Hyg-resistant embryogenic calluses were transferred to MS medium with 1.0 mg L^−1^ ABA and 100 mg L^−1^ Carb, and after 4 weeks of transfer, they formed plantlets (Figure 3D). Nine regenerated plants were transferred to MS medium and seven of them showed dwarf phenotype (Figure 3E). The seven regenerated plants were proved to be transgenic by PCR and GUS analyses, named L1, L2, …, L7, respectively (Figure 3F,G). qRT-PCR analysis revealed that the expression level of *IbNCED1* was significantly increased in most of the transgenic plants compared with that of WT (Figure 3H). 

### 2.4. Plant Height Assay

In vitro propagation of sweet potato is a basic step for routine gene bank and biotechnology research activities. The seven regenerated sweet potato lines were raised in plant numbers by vegetative propagation using an MS medium. The three transgenic sweet potato plants L1, L2, and L4, with high relative expression of *IbNCED1* and stable dwarf phenotype, were selected to test plant height. The result show that overexpression of *IbNCED1* conferred a reduction in height of in vitro-grown and greenhouse-grown transgenic plants (Figure 4A,B). The histological analysis of the longitudinal section showed that the pith cell length of the transgenic plants decreased in comparison to the WT (Figure 4C). All the results demonstrated that *IbNCED1* demoted stem elongation primarily by reducing cell length.

### 2.5. Underlying Mechanism of IbNCED1 in Plant Height

To explore the dwarfing mechanism and the dwarf genes of sweet potato, differentially expressed genes and metabolic pathways in transgenic sweet potato were analyzed by RNA sequencing (RNA-Seq) using 4-week-old in vitro-grown WT and transgenic line L2 (OE). After removing the adapter and low-quality reads, a total of 614,283,286 clean reads were obtained from two lines (three biological replicates per line), and the quality control and quality assessment of RNA-Seq data showed that the sample quality was reliable and can be analyzed later (Appendix A and Appendix A). Using WT as the control group and |log2 (Fold Change)| > 1 & q < 0.05 as the standard of gene differential expression, we obtained a total of 2938 differential expressed genes (DEGs), of which 1827 genes were downregulated, and 1111 genes were upregulated. KEGG enrichment analysis showed that the DEGs were primarily enriched metabolism, biosynthesis of secondary metabolites, plant MAPK signalling, and plant hormone signal transduction pathway (Figure 5A). The DEGs of GA biosynthesis and signal transduction pathway were downregulated (Figure 5B).

To investigate the underlying mechanism of *IbNCED1* in plant height, the phytohormone components of 4−week−old in vitro-grown sweet potato plants were measured. The results showed that the ABA and ABA−GE contents of the transgenic plants were significantly increased, while the GA3 content was significantly decreased compared with those of WT (Table 1). Exogenous GA3 treatment was performed on WT and transgenic sweet potato to determine the factors of height reduction. The WT plant and transgenic sweet potato could not grow on MS with 10 ng L^−1^ GA3 and 30 ng L^−1^ GA3, respectively (Appendix A). These results indicated that overexpression of *IbNCED1* could reduce the GA sensitivity of transgenic sweet potato. To further prove the function of GA in plant height, we analyzed the plant height of transgenic sweet potato plants and WT after GA3 treatment, and the results showed that exogenous GA3 can restore the plant height of transgenic sweet potato (Figure 6). In conclusion, we suggest that *IbNCED1* negatively regulates plant height by controlling the GA biosynthesis and signal transduction pathway.

## 3. Discussion

The Green Revolution has promoted a significant yield increase through the development of semi-dwarf plant architecture in rice, wheat, maize, and soybean [21,44,45,46]. The ideal architecture for sweet potato also could promote the mechanization degree and yield of storage roots. However, the dwarfing mechanism and the dwarf genes of sweet potato are still unclear. In this study, we cloned an *IbNCED1* from the sweet potato cv. Jishu26 (Figure 1). The expression of *IbNCED1* was downregulated in the leaf and upregulated in the stem and root after ABA and GA treatments (Figure 2). Its overexpression significantly conferred a reduction in the height of the transgenic sweet potato plants and promoted the accumulation of ABA and ABA−GE in transgenic sweet potato (Table 1). It is thought that *IbNCED1* is key enzyme gene for ABA biosynthesis signalling in sweet potato.

The other functions of NCED genes have been identified in different plants. *GhNCED1* reduced plant height in cotton [47]. *AtNCED3* and *AtNCED5* contributed to ABA production affecting vegetative growth and drought tolerance in *Arabidopsis* [48,49]. *OsNCED3* mediates seed dormancy, plant growth, abiotic stress tolerance, and leaf senescence by regulating ABA biosynthesis in rice [40]. In our study, overexpression of *IbNCED1* reduced plant height and cell length of the stem in transgenic sweet potato (Figure 4). The results indicated that *IbNCED1* plays an important role in reducing the growth of transgenic sweet potato by regulating ABA biosynthesis.

To date, many Rht genes have been identified in regulating plant height via participating in GA biosynthesis regulation in different plants [11,12,13]. The antagonistic regulations of GA and ABA have been reported in seed germination, cell development of the hypocotyls and plant height [50,51,52,53,54]. The miR528 and its target gene *DWARF3* (D3) negatively regulate rice plant height by triggering a reduction of GA content and a significant increase in ABA accumulation in transgenic plants [55]. Overexpression of *LeNCED1* limited biomass accumulation increased ABA concentration and prevented the induction of genes in ABA metabolism and GA deactivation [42]. GA20−oxidases (GA20oxs) that produce GA precursors, GA3−oxidases (GA3oxs) that produce bioactive GAs, and GA2−oxidases (GA2oxs) that deactivate precursors and bioactive GAs, were kay enzymes of the GA biosynthesis pathway [56,57]. GA20oxs are known to affect cell division and cell expansion, resulting in larger plants [58,59]. GA3ox1 and GA3ox2, which encode a GA3 beta-hydroxylase in GA biosynthesis, are significantly associated with cell lengths and plant height [60,61]. GA2oxs regulate plant growth by regulating endogenous bioactive Gas [62,63]. In the GA signal transduction pathway, the gibberellin receptor GIBBERELLIN INSENSITIVE DWAR (GID) was a putative candidate gene controlling plant height [64,65,66,67]. Interactions between GID1 and DELLAs mediated the GA signalling in land plants [68,69]. The plant-specific gibberellic acid-stimulated *Arabidopsis* (GASA) gene family plays roles in hormone response, promoted seedling germination and root extension, and plant development [70]. In this study, the expression of genes in GA biosynthesis and signal transduction pathways was downregulated in transgenic sweet potato (Figure 5B). Overexpression of *IbNCED1* reduced the accumulation of GA3 and exogenous application of GA3 could rescue the dwarf phenotype (Table 1, Figure 6 and Appendix A). These results suggest that *IbNCED1* regulates plant height and development by controlling the GA signalling pathway in transgenic sweet potato. All the analyses revealed that the occurrence of dwarfing in transgenic sweet potatoes with high ABA content was likely to be caused by the GA signalling pathway.

In wheat and rice, the Rht alleles were introduced to reduce plant height allowing the application of higher fertilizer rates to substantially increase grain yield [11]. The fertilizer rates of the dwarf transgenic sweet potato would impact the yield of storage roots. The main objective of future dwarf sweet potato research should be to optimize fertilizer rates.

## 4. Materials and Methods

### 4.1. Plant Material

Sweet potato cv. Jishu26 was used for isolation and expression analysis of the *IbNCED1* gene. Sweet potato cv. Xushu22 was employed to characterize the function of *IbNCED1*.

### 4.2. Cloning and Sequence Analysis of IbNCED1

Total RNA from sweet potato cv. Jishu26 plants was extracted using the Trozol Up Kit (ET111, Transgen, Beijing, China). The first-strand cDNA was transcribed from the total RNA with the PrimeScript^TM^ RT reagent Kit with gDNA Eraser (PR047A, Takara, Beijing, China). Amino acid sequence alignment was analyzed using DNAMAN V6 software. The phylogenetic tree was constructed with MEGA 7.0 software with 1000 bootstrap replicates. The molecular weight and theoretical isoelectric point (*p*I) of *IbNCED1* were calculated with ProtParam tool (https://web.expasy.org/protparam/) accessed on 9 May 2023.

### 4.3. Expression Analysis of IbNCED1

The transcript levels of *IbNCED1* in leaf, stem and root tissues of the 4-week-old in vitro-grown plants and leaflet, leaf, stem, pencil root, and storage root tissues of the 80-day-old field-grown plants of Jishu26 were analyzed with qRT-PCR using SYBR Green Pro Taq HS kit (AG11701, ACCURATE BIOLOGY). Furthermore, the 4-week-old Jishu26 plants were stressed in Hoagland solution with 100 mM ABA and 100 mM GA, respectively, and sampled at 0, 3, 6, 12, and 24 h after stresses for analyzing the expression of *IbNCED1. Ibactin* (AY905538) was used to normalize the expression levels in sweet potato [71]. All the specific primers are shown in Appendix A.

### 4.4. Regeneration of the Transgenic Sweet Potato Plants

Embryogenic suspension cultures of sweetpotato cv. Xushu22 were prepared using MS medium with 2.0 mg L^−1^ 2,4-D [72]. The overexpression vector pCAMBIA1301-*IbNCED1* was introduced into the *A. tumefaciens* strain EHA105. The transformation and plant regeneration were performed as previously described [70]. The identification of the transgenic plants was conducted by PCR with specific primers (Appendix A). The expression levels of *IbNCED1* in the in vitro-grown transgenic and WT plants were analyzed using specific primers designed in the non-conserved domain (Appendix A).

### 4.5. Plant Height Analysis

The phenotypic of the 4-week-old in vitro-grown transgenic sweet potato plants and WT cultured on MS medium and the 6-week-old plants grown in transplanting boxes in the greenhouse were analyzed. At least 5 plants were measured for plant height. For paraffin section, the stem tissues were collected from WT and transgenic lines. The methods of paraffin section are dissected as described by Fang et al. (2021) [73]. At least 20 cells were measured in length.

### 4.6. RNA-Sequencing and Hormone Analysis

Due to the dwarf phenotype, total RNA was extracted from 4-week-old in vitro-grown sweet potato plant Xushu22 (WT) and transgenic lines L2 (OE) using a plant RNA kit (DP441, TIANGEN). The sequencing library was constructed using Ultra RNA sample preparation kit (Illumina) and then sequenced using an Illumina HiSeq 2500 according to the standard method (Illumina). Total reads were mapped to the *I. Trifida* genome (Sweet potato). Differentially expressed genes were identified using Cuffdiff with default criteria (fold change >1.5) and adjusted false discovery rate (*p* value < 0.05). Three independent biological replicates were used for the RNA-sequencing analysis. Analysis using the Kyoto Encyclopedia of Genes and Genomes (KEGG) pathway was conducted according to database instructions (KEGG PATHWAY Database). The gene expression patterns were graphically represented in a heat map by cluster analysis using TBtools software (version number 1.108). The hormone contents of 4-week-old in vitro-grown WT and transgenic lines L2 plants were determined using high-performance liquid chromatography (HPLC).

### 4.7. Exogenous GA3 Treatment Analysis

In order to investigate the effect of GA3 on plants, the in vitro-grown transgenic and WT plants were cultured on MS medium with 0 (control), 5, 10, 20, 30, and 50 ng L^−1^ GA3 for 4 weeks. Furthermore, we measured the plant height of the in vitro-grown transgenic and WT plants culturing on MS medium with 0 (control) and 10 ng L^−1^ GA3 for 6 weeks.

### 4.8. Statistical Analysis

For cell length, at least 20 biological replicates were analysed. Data were presented as the mean ± SE and analyzed using Student’s *t*-test (two-tailed analysis). For biochemical and molecular biology analysis, all experiments were donefor at least three biological replicates. Significance levels at *p* < 0.05 and *p* < 0.01 are denoted by * (or different small letters) and **, respectively.

## 5. Conclusions

A novel 9-*cis*-epoxycarotenoid dioxygenase gene, *IbNCED1*, was isolated and characterized from sweet potato. Its overexpression in sweet potato led to a semi-dwarf phenotype, increased contents of ABA, decreased levels of GA3, and downregulated gene expression of the GA3 signal transduction pathway. *IbNCED1* overexpression reduced sensitivity to GA3, and exogenous GA3 treatment rescued the dwarfism phenotype. It is suggested that *IbNCED1* regulates plant height by the ABA and GA signalling pathways in transgenic sweet potato.

## Figures and Tables

**Figure 1 ijms-24-10421-f001:**
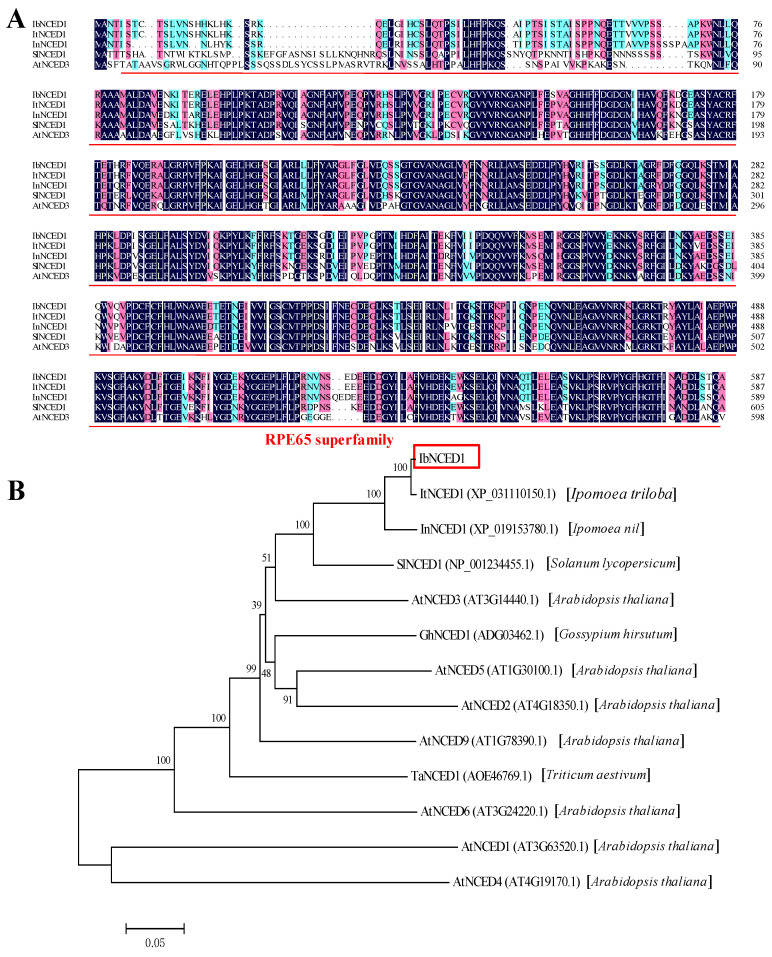
Sequence alignment (**A**) and phylogenetic tree (**B**) of *IbNCED1* with its homologs from other plants.

**Figure 2 ijms-24-10421-f002:**
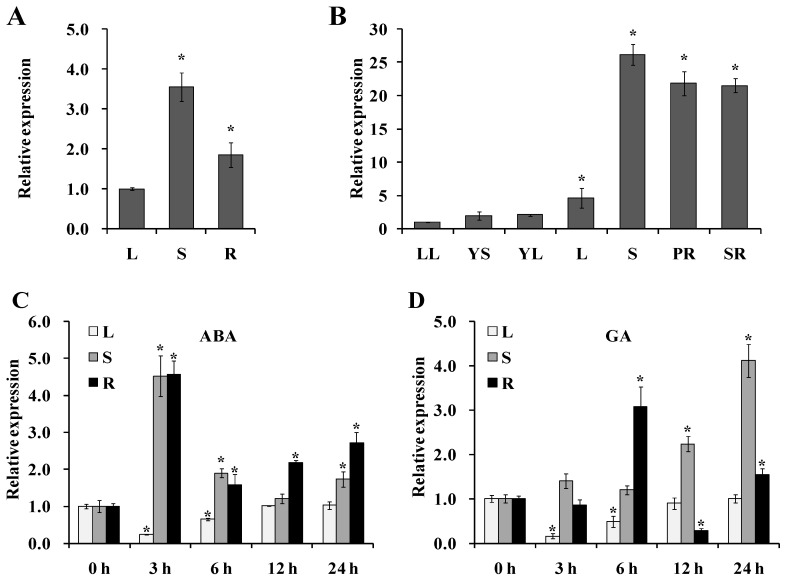
Expression analysis of *IbNCED1* in different tissues of Jishu26 plants. The expression analysis of *IbNCED1* in different tissues of in vitro-grown (**A**) and field-grown (**B**) plants of Jishu26. LL: Leaflet; L: Leaf; PR: Pencil root; R: Root; S: Stem; SR: Storage root; YS: Young stem. The transcript levels of *IbNCED1* in the leaf tissue or leaflet were set to 1. The expression analysis of *IbNCED1* in different tissues of Jishu26 plants after different time points (h) in response to 100 mM ABA (**C**) and 100 mM GA (**D**), respectively. The expression level of *IbNCED1* in the plant sampled at 0 h was set to 1. The data are presented as the means ± SEs (*n* = 3). * indicates significant differences from that of WT at *p* < 0.05, according to Student’s *t*-test.

**Figure 3 ijms-24-10421-f003:**
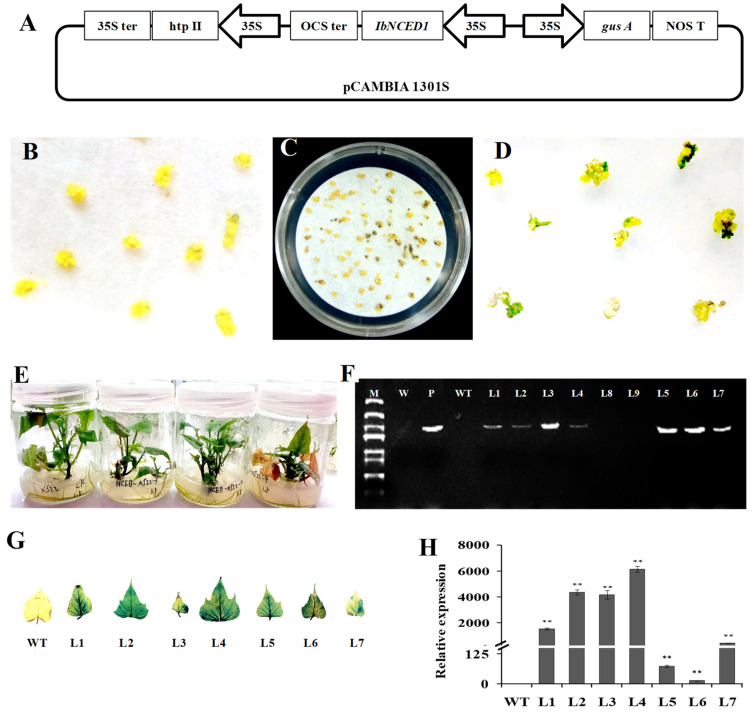
Production of the *IbNCED1*-overexpressing sweet potato plants. (**A**) Diagram of constitutive expression of the 35S promoter::*IbNCED1* construct. (**B**) Embryogenic cultures proliferating in MS medium with 2.0 mg L^−1^ 2,4-D. (**C**) Hyg-resistant calluses formedafter 4 weeks of selection on MS medium with 2.0 mg L^−1^ 2,4-D, 100 mg L^−1^ Carb and 10 mg L^−1^ Hyg. (**D**) Germination of somatic embryos from Hyg-resistant calluses on MS medium with 1.0 mg L^−1^ ABA and 100 mg L^−1^ Carb. (**E**) Whole regenerated plantlets. (**F**) PCR analysis of the transgenic plants. Lane M: BL2000 plus DNA marker; Lane W: Water; Lane P: plasmid pCAMBI1301::*IbNCED1* as a positive control; Lane WT: Xushu22 plant as a negative control. (**G**) GUS analysis of the transgenic plants. (**H**) qRT-PCR analysis of *IbNCED1* in the transgenic plants. ** indicates a significant difference from that of WT at *p* < 0.01 according to Student’s *t*-test.

**Figure 4 ijms-24-10421-f004:**
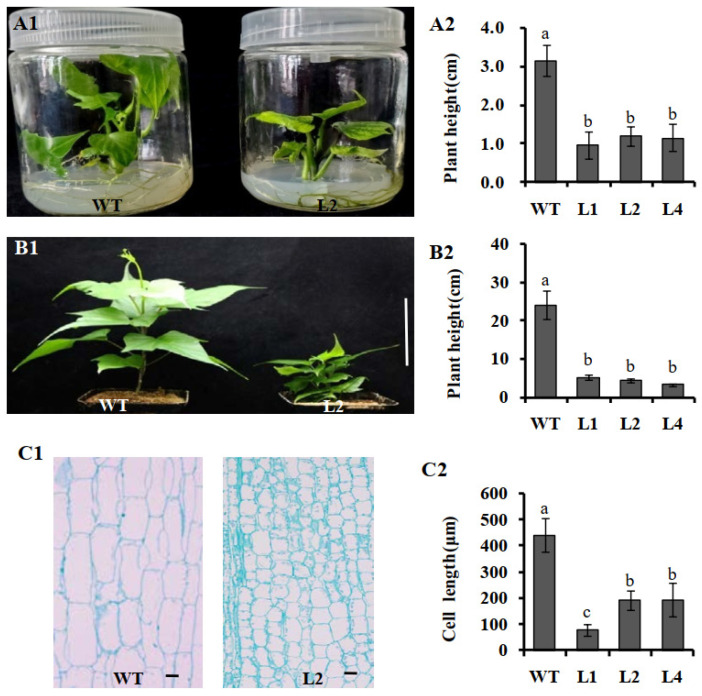
Plant height of transgenic sweet potato plants and WT. (**A1**,**A2**) Phenotypes and plant height of in vitro-grown transgenic sweet potato plants and WT cultured on MS medium for 4 weeks. The data are presented as the means ± SEs (*n* = 5). (**B1**,**B2**) Phenotypes and plant height of transgenic sweet potato plants and WT grown in transplanting boxes for 6 weeks. Bar = 10 cm. The data are presented as the means ± SEs (*n* = 5). (**C1**,**C2**)The histological analysis and cell length of in vitro-grown transgenic sweet potato plants and WT cultured on MS medium for 4 weeks. Bar = 100 μm. The data are presented as the means ± SEs (*n* = 20). The data are presented as the means ± SEs (*n* = 5). The different small letters indicate a significant difference at *p* < 0.05 according to Student’s *t*-test.

**Figure 5 ijms-24-10421-f005:**
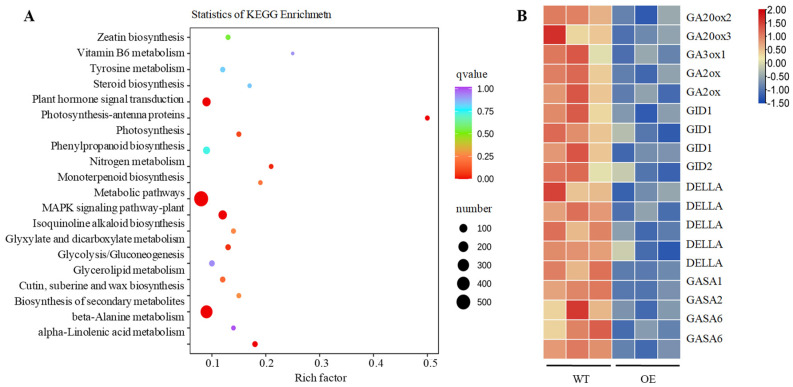
The analysis of KEGG pathway in transgenic sweet potato (**A**) and the expression analysis of DEGs in GA signalling pathway (**B**).

**Figure 6 ijms-24-10421-f006:**
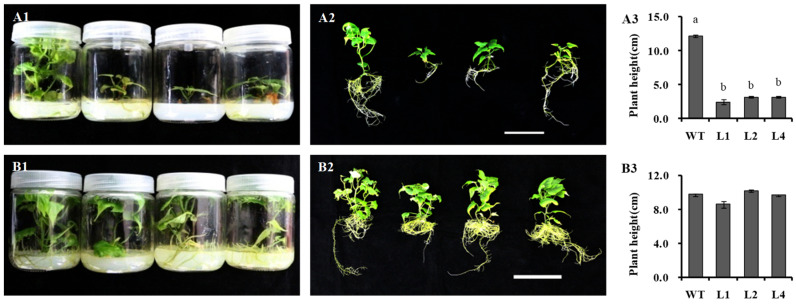
Plant height of transgenic sweet potato plants and WT after GA3 treatment. (**A1**–**A3**) Phenotypes and plant height of in vitro-grown transgenic sweet potato plants and WT cultured on MS medium for 6 weeks. (**B1**–**B3**) Phenotypes and plant height of in vitro-grown transgenic sweet potato plants and WT cultured on MS medium with 10 ng L^−1^ GA3 for 6 weeks. Bar = 10 cm. The data are presented as the means ± SEs (*n* = 3). The different small letters indicate a significant difference at *p* < 0.05 according to Student’s *t*-test.

**Table 1 ijms-24-10421-t001:** The contents of ABA and GAs (mg g^−1^).

Class	Index	WT	L2
ABA	ABA	7.60 ± 0.68	83.42 ± 1.77
	ABA-GE	133.91 ± 4.82	355.59 ± 8.22
GA	GA1	N/A	N/A
	GA3	5.02 ± 0.16	N/A
	GA4	N/A	N/A
	GA7	N/A	N/A
	GA9	N/A	N/A
	GA15	N/A	N/A
	GA19	21.15 ± 0.53	23.85 ± 0.84
	GA20	N/A	N/A
	GA24	N/A	N/A
	GA53	6.36 ±0.68	11.34 ± 0.30

N/A—Not applicable.

## Data Availability

The raw data supporting the conclusions of this article will be made available by the authors, without undue reservation, to any qualified researcher.

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
