# Peer review of "Overexpression of 9-cis-Epoxycarotenoid Dioxygenase Gene, IbNCED1, Negatively Regulates Plant Height in Transgenic Sweet Potato"

_ijms, 2023, doi:10.3390/ijms241310421_

Round 1

Reviewer 1 Report

I have a few comments:

1. There are a lot of type errors in this manuscript, such as in the title  "DioxygenaseGene", it should have a space between the two words. The authors should carefully check grammar/type errors before publication.

2. Fig 3: A, B, C pictures are in low quality, these should be replaced by high resolution figs. For E, the letters are not aligned with the bands. F: it should include all Gus-staining leaf for all 7 lines.

The quality of English is poor, the authors should get professionals polished for this manuscript.

Author Response

Reviewer #1

  1. There are a lot of type errors in this manuscript, such as in the title "Dioxygenase Gene", it should have a space between the two words. The authors should carefully check grammar/type errors before publication.

Revised: Thanks so much for your comments. Based on your comment, we have check grammar/type errors in the new version.

  1. Fig 3: A, B, C pictures are in low quality, these should be replaced by high resolution figs. For E, the letters are not aligned with the bands. F: it should include all Gus-staining leaf for all 7 lines.

Revised: Thanks so much for your comments. Based on your comment, we have replaced Fig 3 by high resolution.

Reviewer 2 Report

In this manuscript the authors report the cloning of a NCED gene from sweet potato. It appears that this is the first NCED gene cloned in this crop, and by using a transgenic approach the authors demonstrated that the overexpression of this gene increases the ABA content and reduces GA signalling, thus impacting in stem elongation and plant height. While the results are exciting and potentially useful to increase crop performance, I think more work is needed to in order to be published.

Major comments:

1-      All the work is based on the phenotyping and analysis of T0 transgenics. Even when this is informative, the proper analyses should be done in stable homozygous T2 plants, where the copy number of insertions have been measured, and if possible, using a null-segregant as a control.  There could be reasons why the authors used in-vitro grown plants (maybe this crop does not produce seeds?), but if so that needs to be explained. In any case insertion copy number and zygosity status are needed.

2-      It is not clear to me if from one primary transgenic, several plants are produced by cloning for the different experiments. Are the plants use for height measurements the same ones used for hormone analysis and hormone treatments?

3-      The assessment of phenotypes in T0, while the plants are still in tissue culture is not ideal. I would better rely on the phenotypes seen in soil.

4-      qPCR expression data needs to be presented as a figure in the main article and not in supplementary materials. Gene expression data in the different transgenics in correlation with the phenotypes of those transgenics needs to be discussed.

5-      A figure with a schematic representation of the construct is needed to easily see the promoter used and is the gene sequence used is the coding region or the full sequenced.

6-      For the transcriptomic analysis is not clear which tissue was used, nor which plant was used and why. Was it a homozygous plant?

7-      Similarly, is not clear which tissue or which plants was used for hormone analysis. And the methods part is missing for that work.

8-      The discussion is too thin and mainly focussed in the GA. In my opinion all the results need discussion, and the authors should point how yield could be impacted in these transgenics. Perhaps they could speculate about future experiments to measure yield, etc.

Minor comments:

2-      The introduction should specifically mention the original dwarfing genes used in wheat for the green revolution, citing original papers or reviews.

3-      In the phylogenetic tree is not clear the species being compared.

4-      The gel with PCR results in Figure 3 needs better alignment of the line names and also negative plants need to be given a number.

Some grammar, missing words and typos need to be checked through the whole manuscript.

Author Response

Reviewer #2

Major comments:

  1. All the work is based on the phenotyping and analysis of T0 transgenics. Even when this is informative, the proper analyses should be done in stable homozygous T2 plants, where the copy number of insertions have been measured, and if possible, using a null-segregant as a control.  There could be reasons why the authors used in-vitro grown plants (maybe this crop does not produce seeds?), but if so that needs to be explained. In any case insertion copy number and zygosity status are needed.

Revised: Thanks so much for your comments.

Sweet potato (Ipomoea batata var. batatas) is an outbreeding, highly heterozygous polyploid (probably hexaploid) crop. It has a complex genome structure and generally shows severe self-incompatibility and keeps heterozygosity within an accession.

As the seeds are heterogeneous, sweet potato is propagated vegetatively. Organs that can be used for the asexual propagation of sweet potato include storage roots, shoot tips, and stem cuttings.

It is one of the most common methods of vegetative propagation raising plants from cuttings for sweet potato.

In many previous studise, regenerated sweet potato plants were proved to be transgenic by three different methods, PCR, GUS analyses and qRT-PCR as described by Liu et al. (2014), Zhang et al. (2019) and Gao et al. (2023).

 Liu, D.G.; Wang, L.J.; Zhai, H.; Song, X.J.; He, S.Z.; Liu, Q.C. A novel alpha/beta-hydrolase gene IbMas enhances salt tolerance in transgenic sweetpotato. PLoS One 2014, 9, e115128, doi:10.1371/journal.pone.0115128.

Zhang, H.; Gao, X.R.; Zhi, Y.H.; Li, X.; Zhang, Q.; Niu, J.B.; Wang, J.; Zhai, H.; Zhao, N.; Li, J.G.; Liu, Q.C.; He, S.Z. A non-tandem CCCH-type zinc-finger protein, IbC3H18, functions as a nuclear transcriptional activator and enhances abiotic stress tolerance in sweet potato. New Phytol. 2019, 223, 1918-1936, doi:10.1111/nph.15925.

Gao, X.R.; Zhang, H.; Li, X.; Bai, Y.W.; Peng, K.; Wang, Z.; Dai, Z.R.; Bian, X.F.; Zhang, Q.; Jia, L.C.; Li, Y.; Liu, Q.C.; Zhai, H.; Gao, S.P.; Zhao, N.; He, S.Z. The B-box transcription factor IbBBX29 regulates leaf development and flavonoid biosynthesis in sweet potato. Plant Physiol. 2023, 191, 496-514, doi:10.1093/plphys/kiac516.

2-      It is not clear to me if from one primary transgenic, several plants are produced by cloning for the different experiments. Are the plants use for height measurements the same ones used for hormone analysis and hormone treatments?

Revised: Thanks so much for your comments. The seven regenerated sweet potato lines were raised plant number by vegetative propagation using MS medium. The different plants of WT, L1, L2 and L4 were used for height measurements, hormone analysis and hormone treatments.

3-      The assessment of phenotypes in T0, while the plants are still in tissue culture is not ideal. I would better rely on the phenotypes seen in soil.

Revised: Thanks so much for your comments. Phenotypes and plant height of transgenic sweet potato plants and WT were measured grown in MS medium or transplanting boxes with soil, respectively (Figure 4).

4-      qPCR expression data needs to be presented as a figure in the main article and not in supplementary materials. Gene expression data in the different transgenics in correlation with the phenotypes of those transgenics needs to be discussed.

Revised: Thanks so much for your comments. Figure S1 has been presented in Figure 2 the revised version.

The relationship of gene expression data in the different transgenics and the phenotypes was be discussed in “2.3. Regeneration of the Transgenic Sweet Potato Plants”.

5-      A figure with a schematic representation of the construct is needed to easily see the promoter used and is the gene sequence used is the coding region or the full sequenced.

Revised: Thanks so much for your comments. A figure with a schematic representation of the construct is presented in Figure 3A.

6-      For the transcriptomic analysis is not clear which tissue was used, nor which plant was used and why. Was it a homozygous plant?

Revised: Thanks so much for your comments. The 4-week-old in vitro-grown sweet potato plant Xushu22 (WT) and transgenic lines L2 were used for the transcriptomic analysis. They were not homozygous plant.

7-      Similarly, is not clear which tissue or which plants was used for hormone analysis. And the methods part is missing for that work.

Revised: Thanks so much for your comments. The methods has been added in the “4.5. RNA-sequencing and hormone analysis”.

8-      The discussion is too thin and mainly focussed in the GA. In my opinion all the results need discussion, and the authors should point how yield could be impacted in these transgenics. Perhaps they could speculate about future experiments to measure yield, etc.

Revised: Thanks so much for your comments. We have added the discussion about yield in these transgenics. The fertilizer rates of the dwarf transgenic sweet potato would impact the yield of storge root. The main objective of the dwarf sweet potato research should optimize the fertilizer rates in the future.

Minor comments:

2-      The introduction should specifically mention the original dwarfing genes used in wheat for the green revolution, citing original papers or reviews.

Revised: Thanks so much for your comments. We have added the dwarfing genes in wheat for green revolution in the introduction part.

3-      In the phylogenetic tree is not clear the species being compared.

Revised: Thanks so much for your comments. Based on your comment, we have replaced Fig 2B with the species beings.

4-      The gel with PCR results in Figure 3 needs better alignment of the line names and also negative plants need to be given a number.

Revised: Thanks so much for your comments. Based on your comment, we have replaced Fig 3 by high resolution.

Round 2

Reviewer 1 Report

No more concerns.

Author Response

Dear reviewer 1,

I am very glad that our manuscript (ID: ijms-2353208) titled “Overexpression of 9-cis-epoxycarotenoid dioxygenase gene, IbNCED1, negatively regulates plant height in transgenic sweet potato” is thought for publication in International Journal of Molecular Sciences.

I greatly appreciate your help for this paper.

Best wishes!

Sincerely Yours,

Yuanyuan Zhou

E-mail: zhou_yy_2020@163.com

Reviewer 2 Report

The authors have reviewed the paper addressing all the major issues. So, I think it can be published, after a minor revision. I recommend:

- In results, add the explanation about why in sweet potato T0 transgenics need to be study.

- Discussion should be more coherent and built around all the results presented in the paper.

- The gel with PCR results in Figure 3 needs better alignment of the line names and also negative plants need to be given a number.

-  In the phylogenetic tree there should not be two IbNCED1 branches. One should be ItNCED1. 

- Not clear which tissue or which plants was used for hormone analysis. 

I detected several grammatical errors or typos that need checking. 

Author Response

Dear reviewer,

Thanks so much for your comments. Based on these comments and suggestions, we have made careful modifications on the original manuscript. All changes made to the text are in red color.

I hope that the revised manuscript is now suitable for publication. Below you will find our point-by-point responses to the reviewers’ comments.

Best wishes!

Sincerely Yours,

Yuanyuan Zhou

(1)- In results, add the explanation about why in sweet potato T0 transgenics need to be study.

Revised: Thanks so much for your comments. We have added the explanation. “In vitro propagation of sweet potato is a basic step for routine genebank and biotechnology research activities. The seven regenerated sweet potato lines were raised plant numbers by vegetative propagation using MS medium. The three transgenic sweet potato plants, L1, L2 and L4, with high relative expression of IbNCED1 and stable drawf phenotype, were selected to test the plant height.”

(2)- Discussion should be more coherent and built around all the results presented in the paper.

Revised: Thanks so much for your comments. We have discussed all the results in the paper.

(3)- The gel with PCR results in Figure 3 needs better alignment of the line names and also negative plants need to be given a number.

Revised: Thanks so much for your comments. We have corrected that mistake in Fig 3.

(4)-  In the phylogenetic tree there should not be two IbNCED1 branches. One should be ItNCED1. 

Revised: Thanks so much for your comments. We have corrected that mistake in Fig 1.

(5)- Not clear which tissue or which plants was used for hormone analysis. 

Revised: Thanks so much for your comments. We have added “4-week-old in vitro-grown sweet potato plants” to explain the question.